# A Multilabel Classification Framework for Approximate Nearest Neighbor Search

**Ville Hyvönen**
Department of Computer Science
University of Helsinki
Helsinki Institute for Information Technology (HIIT)
ville.o.hyvonen@helsinki.fi

**Elias Jääsaari**
Machine Learning Department
Carnegie Mellon University
ejaeaesa@cs.cmu.edu

**Teemu Roos**
Department of Computer Science
University of Helsinki
Helsinki Institute for Information Technology (HIIT)
teemu.roos@cs.helsinki.fi

## Abstract

Both supervised and unsupervised machine learning algorithms have been used to learn partition-based index structures for approximate nearest neighbor (ANN) search. Existing supervised algorithms formulate the learning task as finding a partition in which the nearest neighbors of a training set point belong to the same partition element as the point itself, so that the nearest neighbor candidates can be retrieved by naive lookup or backtracking search. We formulate candidate set selection in ANN search directly as a multilabel classification problem where the labels correspond to the nearest neighbors of the query point, and interpret the partitions as partitioning classifiers for solving this task. Empirical results suggest that the natural classifier based on this interpretation leads to strictly improved performance when combined with any unsupervised or supervised partitioning strategy. We also prove a sufficient condition for consistency of a partitioning classifier for ANN search, and illustrate the result by verifying this condition for chronological $k$-d trees.

## 1 Introduction

Approximate nearest neighbor (ANN) search is a fundamental algorithmic problem. There is a large body of literature on ANN search spanning several research communities, including the machine learning community. Specifically, space-partitioning data structures—such as space-partitioning trees (Friedman et al., 1976; Muja and Lowe, 2014; Dasgupta and Sinha, 2015) and data-dependent hash tables (Indyk and Motwani, 1998; Datar et al., 2004; Weiss et al., 2009)—are machine learning methods commonly used for ANN search.

In this article, we propose an intuitive theoretical framework for partition-based ANN search. In particular, we formulate the candidate set selection directly as a multilabel classification problem where the labels represent the indices of the nearest neighbors of the query point. This formulation suggests that the performance of space-partitioning data structures can be improved by using them in a theoretically justified fashion as partitioning classifiers (Devroye et al., 1996, Chapter 21) instead of searching them under the earlier lookup-based paradigm. Our classification framework also enables applying general purpose classifiers—such as a multilabel random forest—directly as an index structure for ANN search.

36th Conference on Neural Information Processing Systems (NeurIPS 2022).

We start by reviewing the relevant background on ANN search and multilabel classification (Sec. 2), and formulating candidate set selection in ANN search as a multilabel classification task (Sec. 3). In Sec. 4 we define the natural (partitioning) classifier for the general multilabel classification task. In Sec. 6 we show that interpreting the earlier lookup-based candidate set selection methods in the multilabel classification framework of Sec. 3 suggests that they define a suboptimal classifier. Our multilabel formulation also enables us to consider asymptotics in the standard statistical learning framework: we establish a sufficient condition for consistency of a partitioning classifier for ANN search (Sec. 7.1). As a concrete example (Sec. 7.2), we verify this condition for the chronological $k$-d tree (Bentley, 1975) that was the first data structure proposed for accelerating nearest neighbor search. To empirically validate the proposed framework, we show that using a natural classifier that is aligned with the ANN task in conjunction with space-partitioning data structures proposed in the literature leads to strictly improved empirical performance compared to the earlier lookup-based candidate set selection methods (Sec. 8).

## 2    Background and notation

### 2.1    Approximate nearest neighbor search

Let the corpus points $\{c_j\}_{j=1}^m$ and the query point $x$ be vectors in $\mathbb{R}^d$. We call the $k$ corpus points that are closest[1] to the query point $x$ its $k$ *nearest neighbors* and denote the set of their indices by

$$\text{NN}_k(x) := k\!-\!\underset{j=1,\dots,m}{\operatorname{argmin}} \|x - c_j\|, \tag{1}$$

where the notation $k\!-\!\operatorname{argmin} f$ means the set of $k$ values for which the function $f$ has the smallest values, and $\|\cdot\|$ is the Euclidean distance. Other metrics, or—more generally—dissimilarity measures can also be used to define the nearest neighbors.

The trivial solution to the problem of finding the nearest neighbors $\text{NN}_k(x)$ of the query point $x$ is to compute the distances to all the corpus points and sort these distances. However, when the dimensionality of the data is high and the corpus size is large, this brute force solution is often too slow if the application requires fast response times. The first data structure proposed for speeding up nearest neighbor search was the $k$-d tree (Bentley, 1975). However, for high-dimensional data, a $k$-d tree is not faster than the brute force approach for exact nearest neighbor search because of the well-known *curse of dimensionality* that affects the non-parametric statistical methods—including partitioning methods for nearest neighbor search—in general (Lee and Wong, 1977). Although the query speed of index structures for *exact* nearest neighbor search degrades when the dimensionality of the data increases so that they are not an improvement on the brute force approach, this problem can be mitigated by allowing an approximate solution. This is why in modern high-dimensional applications *approximate nearest neighbor* (ANN) search is typically used when a fast solution to the nearest neighbor problem is required.

Algorithms for ANN search can be divided into three categories: graphs (Malkov et al., 2014; Malkov and Yashunin, 2018; Iwasaki and Miyazaki, 2018; Baranchuk et al., 2019), quantization (Jegou et al., 2010; Johnson et al., 2019; Sablayrolles et al., 2019), and space-partitioning methods. In this article, we consider space-partitioning methods that can be further divided into tree-based (Muja and Lowe, 2014; Dasgupta and Sinha, 2015; Jääsaari et al., 2019) and hashing-based (Datar et al., 2004; Aumüller et al., 2019; Gong et al., 2020) algorithms that use trees and hash tables, respectively, as index structures.

Space-partitioning algorithms for ANN search use an index structure to select a *candidate set* $S(x) \subset \{1, \dots, m\}$ of potential nearest neighbors. They then calculate the exact distances between the points in the candidate set and the query point, and return the $k$ nearest points as the approximate nearest neighbors. These algorithms will correctly retrieve a nearest neighbor $j \in \text{NN}_k(x)$ if and only if it belongs to the candidate set. Thus, the *recall* of a space-partitioning algorithm can be written as $\text{Rec}(S(x)) := \frac{1}{k}|\text{NN}_k(x) \cap S(x)|$, where we denote the number of elements of the set $A$ by $|A|$. The performance of an approximate nearest neighbor algorithm is typically measured by its average *recall-query time tradeoff* (see e.g. Aumüller et al. (2019) or Li et al. (2019))—i.e., the average query time required to reach a certain average recall level on a set of test queries.

---

[1]In what follows, we assume that the ties are broken uniformly at random so that the query point always has exactly $k$ nearest neighbors.

## 2.2 Multilabel classification

Consider a standard multi-label classification problem with $m$ labels. Let $X \in \mathbb{R}^d$ be a random variable and let $L(X) \subseteq \{1, \ldots, m\}$ be the corresponding label set. Equivalently, the output variable can be presented in binary encoding by defining $Y \in \{0, 1\}^m$ as an $m$-bit random vector, where

$$Y_j = \begin{cases} 1, & \text{if } j \in L(X), \\ 0 & \text{otherwise.} \end{cases} \tag{2}$$

A multilabel classifier is an $m$-component function $g = (g_1, \ldots, g_m) : \mathbb{R}^d \to \{0, 1\}^m$ that attaches a label set to the value of the input variable $X$. Denote the training set that is assumed to be an i.i.d. sample from the distribution of the pair $(X, Y)$ by $D_n := \{(X_i, Y_i)\}_{i=1}^n$. When the classifier $g : \mathbb{R}^d \times \{\mathbb{R}^d \times \{0, 1\}^m\}^n \to \{0, 1\}^m$ is learned from the training set of size $n$, we denote it by $g^{(n)}(x) := g(x, D_n)$. When the training set $D_n$ is considered a random variable, the classifier $g^{(n)}$ also becomes a random function.

The performance of the classifier is measured by a loss function $L : \{0, 1\}^m \times \{0, 1\}^m \to \mathbb{R}$, and the objective is to minimize the risk $\mathcal{R}(g) := E[L(g(X), Y)]$. This risk is lower-bounded by the *Bayes risk* $\mathcal{R}^* = \inf_g \mathcal{R}(g)$, the minimizer of which is called the *Bayes classifier*.

The Bayes classifier for many common multilabel loss functions—such as Hamming loss, ranking loss, precision, recall, and $F$-measures—is obtained by thresholding the conditional label probabilities $\eta_j(x) := P\{Y_j = 1 \mid X = x\}$ (Dembczynski et al., 2010; Koyejo et al., 2015). This justifies the standard plug-in approach of first estimating the conditional label probabilities[2] by $\hat{\eta}_1(x), \ldots, \hat{\eta}_m(x)$, and then defining the *plug-in classifier* as

$$g_j^{(n)}(x) := \begin{cases} 1, & \text{if } \hat{\eta}_j(x) > \tau \\ 0, & \text{otherwise,} \end{cases} \tag{3}$$

where $\tau \in [0, 1]$; equivalently, the plug-in classifier can be written as the estimate of the label set $L(x)$ as $\hat{L}(x) := \{j \in \{1, \ldots, m\} : \hat{\eta}_j(x) > \tau\}$.

The multilabel classification problem is often solved by reducing it to a series of binary or multiclass classification problems, and estimating the conditional label probabilities $\eta_j(x)$ under this model. (see, e.g., Menon et al. (2019) for a discussion of different reduction methods). In what follows, we will employ the *pick-all-labels* (PAL) reduction (Reddi et al., 2019) where we separate each label $l \in L(x_i)$ of the training set point $x_i$ into a multi*class* (but single-label) training instance $(x_i, l)$, and fit the classifier to this modified training set by minimizing a multiclass loss function.

## 3 Candidate set selection as a multilabel classification problem

Equipped with the above definitions, we are now in a position to formalize candidate set selection in ANN search described in Sec. 2.1 as an instance of the multilabel classification problem described in Sec. 2.2.

In the classical formulation of ANN search, the input–output pair is defined as $(x, \text{NN}_k(x))$. It is straightforward to observe that this is essentially an instance of the multilabel classification problem where $\text{NN}_k(x)$—i.e., the set of indices of the $k$ nearest neighbors of the query point—is the label set $L(x)$. Assuming that the values of $x$ are i.i.d. draws from the distribution of the random variable $X$ (the *query distribution*), the objective is to predict the value of the random variable $Y$ (defined by (2) with $\text{NN}_k(X)$ as a label set) given the value of the random variable $X$. A distinctive property of this classification problem, which follows from the definition of the ANN task, is that the size of the label set $|L(x)| = \sum_{j=1}^m y_j = k$ is constant for all queries.

Since the labels $\{1, \ldots, m\}$ correspond to the indices of the corpus points, the classification decision (3) where the probability estimates are thresholded corresponds to candidate set selection, and the estimated label set $\hat{L}(x)$ corresponds to the candidate set $S(x)$.

---

[2]More generally, instead of the conditional label probability estimates $\hat{\eta}_1(x), \ldots, \hat{\eta}_m(x)$, any score function values $s_1(x), \ldots, s_m(x)$ for the labels can be learned and thresholded to make the classification decision. While we will present all the results only for the version of the plug-in classifier that uses the probability estimates, they readily generalize to the version of the plug-in classifier that uses the score function values.

If no additional training data is available, the corpus itself can be used as a training set. More precisely, in this case we interpret $\{c_j\}_{j=1}^m$ as a sample from the query distribution, compute the $k$ nearest neighbors of the corpus points, and then use $\{(c_j, y_j)\}_{j=1}^m$ as a training set. Note that in this case $y_{jj} = 1$ for each $j = 1, \ldots, m$ since each corpus point is the nearest neighbor of itself.

# 4 Partitioning classifiers

In this section, we first give a general definition of the partitioning classifier. We then define the *natural classifier*—that is a special case of the partitioning classifier—for the single partition and for the ensemble of partitions for the general multilabel classification problem described in Sec. 2.2.

*Partitioning classifier* is a general term for a classifier that is based on learning a partition of the instance space and whose classification decision is based on the labels of the training set points that belong to the same partition element as the query point. Partitioning classifiers can be divided into two categories depending on whether the partition is flat or recursive. There is a vast literature on recursive partitioning classifiers (i.e., classification trees), and gradient boosted trees (Friedman et al., 2000; Friedman, 2001) are one of the most widely used and efficient classifiers (Chen and Guestrin, 2016). Flat partitions are more typically used for density estimation (Kontkanen and Myllymäki, 2007; López-Rubio, 2013; Cui et al., 2021), but they have also been used for classification (Lugosi and Nobel, 1996; McAllester and Ortiz, 2003).

Denote by $\mathcal{P} = \{R_1, R_2, \ldots, R_L\}$ the partition of $\mathbb{R}^d$, i.e., a collection of disjoint sets for which $\bigcup_{l=1}^L R_l = \mathbb{R}^d$. Denote the structure function that maps the query point to the index of the partition element it belongs into by $q : \mathbb{R}^d \to \{1, 2, \ldots, L\}$. When the partition is learned from the training data, we denote it by $\mathcal{P}^{(n)} = \pi(D_n)$, where $\pi(D_n)$ is a partitioning rule that associates the training set with a partition of $\mathbb{R}^d$.

**Natural classifier for a single partition.** Partitioning classifiers use the training set twice: first, to learn the partition $\mathcal{P}^{(n)} = \pi(D_n)$, and second, to classify the query point using the training set points that belong to the same partition element $R_{q(x)}$ as the query point $x$. In the multiclass classification, the conditional label probabilities can be estimated in a natural fashion by the observed label proportions

$$\hat{\eta}_j(x) = \frac{1}{N_{q(x)}} \sum_{i \,:\, x_i \in R_{q(x)}} y_{ij}, \tag{4}$$

where $N_{q(x)} := |\{i \,:\, x_i \in R_{q(x)}\}|$ denotes the number of training set points in the same partition element as the query point. This standard practice can be motivated by noting that these observed label proportions are the maximum likelihood estimates of the piecewise constant multinomial model where the conditional label probabilities are constants at each of the partition elements.

In the multilabel case, the estimation of the conditional label probabilities by (4) can be motivated via the PAL reduction under which the observed label proportions in $R_{q(x)}$ are proportional to the maximum likelihood estimates of the piecewise constant multinomial model. To classify the query point, the conditional probability estimates (4) are plugged into (3), i.e., the query point is assigned into all the classes whose probability estimate is greater than or equal to the value of the threshold parameter $\tau$. We call this partitioning classifier the *natural classifier*.

**Natural classifier for an ensemble of partitions.** When a collection of partitions $\{\mathcal{P}_t^{(n)}\}_{t=1}^T$, where $\mathcal{P}_t^{(n)} := \{R_1^{(t)}, \ldots, R_{L_t}^{(t)}\}$, is used as a classifier—such as in random forests (Ho, 1998; Breiman, 2001)—the contributions of the partitions are aggregated. In this article, we consider the most straightforward aggregation method where the conditional probability estimates are obtained as averages of the conditional probability estimates of the individual partitions:

$$\hat{\eta}_j(x) = \frac{1}{T} \sum_{t=1}^T \hat{\eta}_j^{(t)}(x). \tag{5}$$

The estimate of the single partition $\hat{\eta}_j^{(t)}(x)$ is defined as in (4) for the corresponding partition $\mathcal{P}_t^{(n)}$ and the corresponding structure function $q_t$.

# 5    Related work

The most directly relevant earlier literature consists of studies that learn space-partitioning index structures for ANN search using supervised information. The idea of optimising the index structure for the particular query distribution was first presented by Maneewongvatana and Mount (2001), and later extended by Cayton and Dasgupta (2008) who formulate ANN search as a supervised learning problem and propose a tree-based and a hashing-based algorithm for solving it. More recently, many supervised *learning to hash*-methods, such as minimal loss hashing (Norouzi and Fleet, 2011), LDA hashing (Strecha et al., 2011), and kernel-based supervised hashing (Liu et al., 2011), have also been proposed for ANN search (see, e.g., Wang et al. (2015) or Wang et al. (2017) for a survey).

However, the earlier supervised methods pose the supervised learning problem in an indirect fashion. This is because they, like the earlier unsupervised methods, select the candidate set using a method which we call *lookup search*[3]: they select the corpus point into the candidate set if and only if it belongs to the same partition element as the query point. Consequently, their objective is to learn a partition in which the $k$ nearest neighbors of a query point belong to the same partition element with it. In contrast, our objective is to directly learn a partitioning classifier that predicts its nearest neighbors correctly. We will elucidate the difference in the next section.

# 6    Candidate set selection for ANN search

In view of the multilabel formulation of Sec. 3, the natural classifier defined in Sec. 4 can directly be used to select a candidate set for ANN search. However, in this section we interpret also the earlier lookup-based candidate set selection methods (lookup search and voting search) in our multilabel classification framework, and show that they define a classifier of a different form. We show that this classifier—that we call the *naive classifier*—is in fact the natural classifier for the different multilabel classification problem where the labels do not represent the corpus points that are the nearest neighbors of the query point but the corpus points that belong to the same partition element as the query point. This suggests that the natural classifier would be a more suitable candidate set selection method for ANN search than the naive classifier. The empirical results of Sec. 8 indicate that this is indeed the case.

**Candidate set selection for a single partition.**    First, assume that we utilize the single fixed partition $\mathcal{P} = \{R_1, \ldots, R_L\}$ and the training set $\{(x_i, y_i)\}_{i=1}^n$ to approximate the nearest neighbors of the query point $x$. The natural classifier defined in Sec. 4 selects the candidate set as

$$\hat{L}(x) = \{j \in \{1, \ldots, m\} \,|\, \hat{\eta}_j(x) > \tau\}, \tag{6}$$

where $\tau \in [0, 1]$, and the conditional label probability estimates $\hat{\eta}_j(x) = \frac{1}{N_{q(x)}} \sum_{i\,:\,x_i \in R_{q(x)}} y_{ij}$ are obtained as the observed label proportions among the training set points that belong to the same partition element with the query point. In contrast, lookup search selects the candidate set as

$$\hat{L}(x) = \{j \in \{1, \ldots, m\} \,|\, c_j \in R_{q(x)}\}, \tag{7}$$

i.e., it selects the corpus point into the candidate set if and only if it belongs to the same partition element with the query point. When interpreted in the classification framework of Sec. 3, (7) defines the classifier $\hat{L}(x) = \{j \in \{1, \ldots, m\} \,|\, \tilde{\eta}_j(x) > \tau\}$, where $\tau \in [0, 1)$ and $\tilde{\eta}_j(x) = \mathbb{1}_{R_{q(x)}}(c_j)$; we call this a *naive classifier*.

We immediately observe that the naive classifier is not a natural classifier for the multilabel classification problem in which the labels are defined as $y_{ij} = \mathbb{1}_{\mathrm{NN}_k(x_i)}(c_j)$ as in Sec. 3. Instead, it is a natural classifier for the different multilabel classification problem in which the labels are defined

---

[3]Often, lookup search is combined with a priority queue guided backtracking search in which the query point is routed into more than one element in a partition, and all the corpus points that belong to these partition elements are then selected into the candidate set. This technique is called *priority search* (Arya and Mount, 1993; Silpa-Anan and Hartley, 2008) or *multi-probe LSH* (Lv et al., 2007), depending on whether the index structure is tree-based or hashing-based, respectively. For clarity, we do not consider backtracking—that is mainly a memory-saving technique—in the analysis below: it can be easily incorporated both into lookup search and our method by allowing the structure function $q$ to return a set of indices instead of a single index, and considering $\bigcup_{l \in q(x)} R_l$ instead of the single partition element $R_{q(x)}$.

as $\tilde{y}_{ij} = \mathbb{1}_{R_q(x_i)}(c_j)$ . In other words, the naive classifier is geared towards the learning problem in which—instead of the $k$ nearest neighbors of the query point—the labels represent the corpus points that belong to the same partition element as the query point. The candidate set selection method (7) also explains why the objective of the earlier supervised methods for *learning* the partition differs from ours: in these methods, the objective is to maximise the number of nearest neighbors of the query point that belong to the same partition element with it in order to maximise the recall (while minimising the number of non-neighbors in that element in order to maximise precision).

**Candidate set selection for an ensemble of partitions.** Assume that the fixed set of partitions $\{\mathcal{P}_t\}_{t=1}^{T}$ is used to approximate the $k$ nearest neighbors of a query point $x$. The natural classifier defined in Sec. 4 selects the candidate set

$$\hat{L}(x) = \{j \in \{1, \ldots, m\} \,|\, \hat{\eta}_j(x) > \tau\}, \tag{8}$$

as in (6), but now $\hat{\eta}_j(x) = \frac{1}{T}\sum_{t=1}^{T}\hat{\eta}_j^{(t)}(x)$, where the contributions of the individual partitions $\hat{\eta}_j^{(t)}(x)$ are defined as above. In contrast, the earlier (both supervised and unsupervised) methods select the corpus point into the candidate set if and only if it belongs to the same partition element as the query point in at least one of the $T$ partitions. Hence, the candidate set selected by lookup search is

$$\hat{L}(x) = \left\{ j \in \{1, \ldots, m\} \,\Big|\, c_j \in \bigcup_{t=1}^{T} R_{q^{(t)}(x)}^{(t)} \right\} = \{j \in \{1, \ldots, m\} \,|\, \tilde{\eta}_j(x) > \tau\}, \tag{9}$$

where $\tilde{\eta}_j(x) := \frac{1}{T}\sum_{t=1}^{T}\tilde{\eta}_j^{(t)}(x)$, $\tau \in [0, \frac{1}{T})$, and the contributions of the partitions $\hat{\eta}_j^{(t)}(x)$ are defined as above.

Unlike in the case of a single partition—where the value of the threshold parameter $\tau \in [0, 1)$ does not affect the classification decision of the naive classifier, since $\tilde{\eta}_j(x) \in \{0, 1\}$—now $\tau$ affects the classification decision, since $\tilde{\eta}_j(x) \in \{0, \frac{1}{T}, \ldots, \frac{T-1}{T}, 1\}$. Hence, a tuning parameter can be added to lookup search by allowing $\tau$ to be chosen freely as proposed by Hyvönen et al. (2016) who call the resulting method *voting search*.

## 7 Consistency of partitioning classifiers for ANN search

The ideal index structure for ANN search always returns a candidate set that contains all the $k$ nearest neighbors of the query point and no other corpus points. Under the multilabel formulation, this corresponds to a classifier for which the expected multilabel 0-1 loss $EL(g(X), Y) = P\{g(X) \neq Y\}$ is zero. To this end, we prove a sufficient condition for the consistency of a partitioning classifier for ANN search under 0-1 loss. Consistency under 0-1 loss also directly implies consistency for the other common multilabel loss functions, such as Hamming loss, precision, recall, and $F$-measures. As a concrete example, we prove the consistency of the chronological $k$-d tree (Bentley, 1975) by checking that this condition holds for it.

### 7.1 Sufficient condition for consistency

The classical theorem for proving consistency of partitioning classifiers for binary classification is:

**Theorem 1.** *(Devroye et al. (1996), Theorem 6.1, p. 94–95) Assume that only the features $X_1, \ldots, X_n$ are used to learn the partition $\mathcal{P}^{(n)} = \pi(X_1, \ldots, X_n)$. The natural classifier[4] $g^{(n)}$ defined by $\mathcal{P}^{(n)}$ is consistent (under 0-1 loss) for binary classification, if*

- *(i) $N_{q(X)} \to \infty$ in probability, and*

- *(ii) $\mathrm{diam}\left(R_{q(X)}\right) \to 0$ in probability,*

*when $n \to \infty$.*

---

[4]The natural classifier for binary classification is defined as the classifier that classifies the query point into the majority class of the training set points that belong to the same partition element with it.

The number of the training set points in the partition element the query point $x$ belongs to is denoted by $N_q(x) := |\{i : X_i \in R_{q(x)}\}|$, and the diameter of a set $A$ is defined as the maximum distance between any two points of this set and denoted by $\operatorname{diam}(A) := \sup_{a, b \in A} \|a - b\|$.

While this result is for binary classification, it can be readily extended to the multilabel case. However, as a multilabel classification problem, ANN search has two distinguishing properties: ($i$) the Bayes error $\mathcal{R}^*$ is zero; ($ii$) decision boundaries between the labels consist of subsets of hyperplanes. It turns out that in this case, the second condition of Theorem 1 is sufficient for the consistency of a partitioning classifier:

**Theorem 2.** *Let $g^{(n)}$ be a natural classifier defined by the partition $\mathcal{P}^{(n)} = (R_1, \ldots, R_L)$ and the threshold parameter $\tau \in [0, 1)$ for ANN search. Assume that the distribution of $X$, denoted by $\mu$, is continuous. If $\operatorname{diam}(R_{q(X)}) \to 0$ in probability—that is, if for every $\epsilon > 0$,*

$$P\{\operatorname{diam}(R_{q(X)}) > \epsilon\} \to 0$$

*when $n \to \infty$, then the classifier $g^{(n)}$ is consistent (for 0-1 loss)—i.e., $E_{D_n}\mathcal{R}(g^{(n)}) \to 0$.*

*Proof.* If for all the pairs of corpus points $(c_j, c_{j'})$, $j' \neq j$, all the points of the partition element $R_l$ are closer to $c_j$ than $c_{j'}$ (or vice versa)—that is, if there is no such pair $(c_j, c_{j'})$ for which there exists $a, b \in R_l$ such that $\|a - c_j\| < \|a - c_{j'}\|$ and $\|b - c_j\| > \|b - c_{j'}\|$—then also $\hat{\eta}_j(x) = \eta_j(x)$ for each $x \in R_l$ and $j = 1, \ldots, m$; consequently, each $x \in R_l$ is classified correctly for any $\tau \in [0, 1)$. Now, since for each $j = 1, \ldots, m$,

$$P\{g_j^{(n)}(X) \neq \eta_j(X)\}$$
$$\leq P\left(\exists j' \neq j : \exists a, b \in R_{q(X)} \text{ s.t. } \|a - c_j\| < \|a - c_{j'}\|, \|b - c_j\| > \|b - c_{j'}\|\right)$$
$$\leq \sum_{j' \neq j} P\{\exists a, b \in R_{q(X)} \text{ s.t. } \|a - c_j\| < \|a - c_{j'}\|, \|b - c_j\| > \|b - c_{j'}\|\},$$

to prove consistency of $g^{(n)}$ it is sufficient to show that for all $j, j' \in \{1, \ldots, m\}, j \neq j'$,

$$P\{\exists a, b \in R_{q(X)} \text{ s.t. } \|a - c_j\| < \|a - c_{j'}\|, \|b - c_j\| > \|b - c_{j'}\|\} \to 0$$

in probability when $n \to \infty$.

Choose any $j, j', j \neq j'$, and denote the hyperplane that is halfway in between the corpus points $c_j$ and $c_{j'}$ by $H := \{x \in \mathbb{R}^d : \|x - c_j\| = \|x - c_{j'}\|\}$. For any $t = 1, 2, \ldots$, let $H_t$ denote the set surrounding $H$ by a margin of width $1/t$. Since $H_1 \supset H_2 \supset H_3 \ldots$, and $H = \cap_{t=1}^{\infty} H_t$, it follows from the upper continuity of the probability measure that $\lim_{t \to \infty} \mu(H_t) = \mu(H)$. Because the Lebesgue measure of the hyperplane $H$ in $\mathbb{R}^d$ is zero and $\mu$ is absolutely continuous w.r.t. the Lebesgue measure by the assumption, then also $\lim_{t \to \infty} \mu(H_t) = \mu(H) = 0$.

Now, for any $t = 1, 2, \ldots$, if $R_{q(x)}$ crosses the hyperplane $H$, then either $x \in H_t$ or the diameter of the $R_{q(x)}$ is greater than $1/t$. Hence,

$$P\{\exists a, b \in R_{q(X)} \text{ s.t. } \|a - c_j\| < \|a - c_{j'}\|, \|b - c_j\| > \|b - c_{j'}\|\}$$
$$\leq P\{X \in H_t \text{ or } \operatorname{diam}(R_{q(X)}) > 1/t\}$$
$$\leq \mu(H_t) + P\{\operatorname{diam}(R_{q(X)}) > 1/t\}.$$

We can get $\mu(H_t)$ as small as desired by choosing a large enough $t$; and since by assumption the second term is arbitrarily small when $n$ is large enough, the result follows. $\square$

## 7.2 Consistency of chronological k-d tree

Next, we illustrate the utility of Theorem 2 by applying it to prove the consistency of the *chronological k-d tree* (Bentley, 1975) that rotates the split directions and uses the same split direction for all the nodes at one level of a tree. At the first level the training data is split at the median of the first coordinates of the data points. At the second level both nodes are split at the median of the second coordinates of the node points. At the $(d+1)$th level, the nodes are split again at the median of the first coordinates, and so on (see Appendix C.1).

More precisely, let $X, X_1, \ldots, X_n \in \mathbb{R}^d$ be i.i.d. random variables. A chronological $k$-d tree can be formalized as a partitioning rule $\pi$ that returns the partition $\mathcal{P}^{(n)} = \pi(X_1, \ldots, X_n)$. When the tree height is $\ell$, this partition has $2^\ell$ elements (also called *leafs*). The leafs are hyperrectangles in $\mathbb{R}^d$. Some of the edges of these hyperrectangles may have an infinite length. To handle these leafs, we introduce the notation where, for any $M > 0$, the hypercube $[-M, M]^d$ divides the partition elements $R_1, \ldots, R_{2^\ell}$ into three disjoint sets:

$$
\begin{aligned}
A &:= \{l \in \{1, \ldots, 2^\ell\} \,:\, R_l \subset [-M, M]^d\}, \\
C &:= \{l \in \{1, \ldots, 2^\ell\} \,:\, R_l \subset \mathbb{R}^d \setminus [-M, M]^d\}, \\
B &:= \{1, \ldots, 2^\ell\} \setminus (A \cup C).
\end{aligned}
\tag{10}
$$

Here $A$ is the set of indexes of the partition elements that are completely inside the hypercube $[-M, M]^d$, $B$ is the set of indexes of the partition elements that cross its boundary, and $C$ is the set of indexes of the partition elements that are completely outside of it.

First, we prove two auxiliary results that bound the number of nodes crossing the boundary of the box $[-M, M]^d$ and the combined length of the edges (in any fixed coordinate direction) of the nodes that reside completely inside $[-M, M]^d$, respectively. Note that these bounds are of purely combinatorial nature and thus do not depend on the training set. The proofs of the following results are presented in Appendix A.

**Lemma 1.** *For any training set $D_n$, it holds for the number of nodes of a chronological $k$-d tree—denoted by $N_B := |B|$—crossing the border of the hypercube $[-M, M]^d$ that*

$$
N_B \leq 4d \cdot 2^{\ell - \frac{\ell}{d}}.
$$

**Lemma 2.** *Let $j \in \{1, \ldots, d\}$ be any coordinate direction. Denote the length of the node $R_l$ in the $j$th coordinate direction by $V_l$. Then for any training set $D_n$,*

$$
\sum_{l \in A} V_l \leq 4M \cdot 2^{\ell - \frac{\ell}{d}}.
$$

We are now in a position to establish the consistency of the chronological $k$-d tree for approximate nearest neighbor search. In view of Theorem 2 it suffices to prove that the leaf diameter converges to zero in probability:

**Theorem 3.** *If for the height of a chronological $k$-d tree holds that $\ell \to \infty$ when $n \to \infty$, then the leaf diameter $\mathrm{diam}(R_{q(X)})$ converges to zero in probability.*

## 8  Experiments

We present empirical results validating the utility of our framework. In particular, we compare the natural classifier to the earlier candidate set selection methods discussed in Sec. 6 for different types of unsupervised trees that have been widely used for ANN search. Specifically, we use ensembles of randomized $k$-d trees (Friedman et al., 1976; Silpa-Anan and Hartley, 2008), random projection (RP) trees (Dasgupta and Freund, 2008; Hyvönen et al., 2016), and principal component (PCA) trees (Sproull, 1991; Jääsaari et al., 2019) (see Appendix C for detailed descriptions of these data structures). Another consequence of the multilabel formulation of Sec. 3 is that it enables using any established multilabel classifier for ANN search. To demonstrate this concretely, we train a random forest consisting of standard multilabel classification trees (trained under the PAL reduction (Reddi et al., 2019) by using multinomial log-likelihood as a split criterion) and use it as an index structure for ANN search; it turns out that the fully supervised classification trees have an improved performance compared to the earlier unsupervised trees on some—but, curiously, not on all—data sets.

We follow a standard ANN search performance evaluation setting (Aumüller et al., 2019; Li et al., 2019) by using the corpus as the training set, searching for $k = 10$ nearest neighbors in Euclidean distance, and measuring performance by evaluating average recall and query time over the test set of 1000 points. We use four benchmark data sets: Fashion ($m = 60000$, $d = 784$), GIST ($m = 1000000$, $d = 960$), Trevi ($m = 101120$, $d = 4096$), and STL-10 ($m = 100000$, $d = 9216$). All the algorithms are implemented in C++ and run using a single thread. We tune the hyperparameters

by grid search and plot the Pareto frontiers of the optimal hyperparameters. Further details of the experimental setup are found in Appendix B. The code used to produce the experimental results is attached as supplementary material and can also be found at `https://github.com/vioshyvo/a-multilabel-classification-framework`.

**Comparison of candidate set selection methods.** The candidate set selection method proposed in this article is the natural classifier (8) described in Sec. 4; for completeness, we also include the special case obtained by fixing $\tau = 0$ in the comparison. The earlier methods are lookup search (naive classifier (9) with $\tau = 0$) and voting (Hyvönen et al., 2016; Jääsaari et al., 2019) (naive classifier (9) with $\tau$ as a free tuning parameter). The results for the Trevi data set are presented in Fig. 1 and indicate, as the discussion of Sec. 6 suggests, that the natural classifier performs better than the earlier lookup-based methods for all types of trees (this finding holds consistently over all the data sets in our experiments; see Fig. 2 in Appendix).

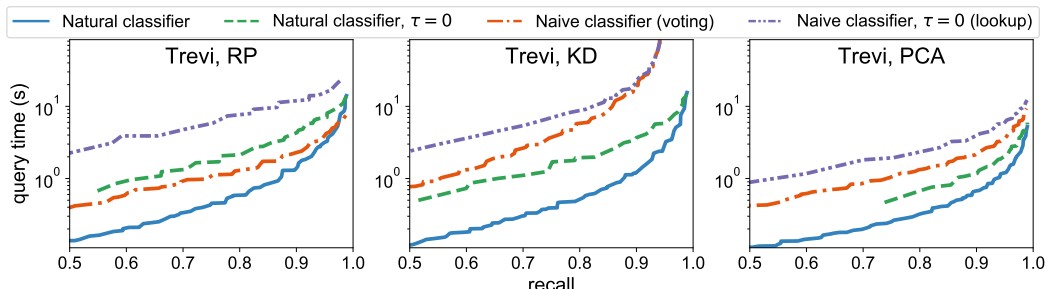

Figure 1: Recall vs. query time (log scale) of ensembles of, RP, $k$-d, and PCA trees. The solid blue line is the natural classifier proposed in this paper; the dash-dotted red line is the natural classifier with $\tau = 0$ that is included for completeness; the dashed green line is voting; and the double-dash-dotted violet line is lookup search. The natural classifier is the fastest and the lookup search is the slowest of the methods for each tree type.

**Comparison of tree types.** We compare the aforementioned ensembles of unsupervised (RP, KD, and PCA) trees and the random forest consisting of supervised classification trees (RF); for all four tree types the candidate set is selected by (8). The results are shown in Table 1. Since the random forest (RF) leverages supervised information to learn the trees, we would expect that it is the fastest tree-based method. Indeed, this is the case on Fashion and GIST. However, on STL-10 and Trevi, the unsupervised PCA tree is the fastest method. We hypothesize that this is because of the high dimensionality of STL-10 and Trevi: standard supervised classification trees employed by random forest are restricted to axis-aligned splits, whereas PCA trees—although they use an unsupervised split criterion—can find more informative oblique split directions. An interesting topic for future work would be to apply supervised classification trees that can utilize oblique split directions.

## 9 Conclusion

We establish a general theoretical framework for ANN search by formulating candidate set selection as a multilabel learning task. Empirical results validate our framework: a natural classifier derived directly from the problem formulation is a strict improvement over the earlier lookup-based candidate set selection methods. In addition, we provide a sufficient condition that guarantees consistency of a partitioning classifier for ANN search. We verify this condition for chronological $k$-d trees, indicating that—given enough training data—they retrieve a candidate set containing all the $k$ nearest neighbors of the query point and no other corpus points.

**Limitations.** Supervised ANN search methods typically have longer pre-processing times compared to unsupervised methods. This is because (1) they require computing the true nearest neighbors $\{y_i\}_{i=1}^n$ of the training set points $\{x_i\}_{i=1}^n$ and (2) supervised index structures are often slower to build compared to their unsupervised counterparts (c.f. Appendix D.1). If fast index construction is required, the second problem can be mitigated by learning trees in an unsupervised fashion, but using

Table 1: Query times (seconds / 1000 queries) at different recall levels for the different tree types. The fastest method in each case is typeset in boldface.

| data set | R (%) | PCA | KD | RP | RF |
|---|---|---|---|---|---|
| Fashion | 80 | 0.075 | 0.076 | 0.099 | **0.063** |
| | 90 | 0.111 | 0.126 | 0.172 | **0.095** |
| | 95 | 0.163 | 0.171 | 0.261 | **0.146** |
| GIST | 80 | 1.330 | 0.958 | 1.009 | **0.705** |
| | 90 | 2.942 | 2.286 | 2.226 | **1.530** |
| | 95 | 5.641 | 4.451 | 4.598 | **3.253** |
| STL-10 | 80 | **0.382** | 0.872 | 1.211 | 0.756 |
| | 90 | **0.756** | 2.126 | 3.248 | 1.774 |
| | 95 | **1.315** | 4.376 | 7.330 | 3.654 |
| Trevi | 80 | **0.330** | 0.543 | 0.591 | 0.582 |
| | 90 | **0.684** | 1.464 | 1.468 | 1.234 |
| | 95 | **1.212** | 3.244 | 3.289 | 2.350 |

them as partitioning classifiers as described in Sec. 4, since the experiments of Sec. 8 suggest that the candidate set selection method has a more pronounced effect on the performance than the tree type.

**Future research directions.** While we demonstrate our approach using a random forest classifier, we expect that the most important consequence of our work is that it enables using any type of classifier as an index structure for ANN search. In particular, gradient boosted trees (Friedman, 2001) are promising since they are often more accurate than random forests. *Extreme classification* models, including tree-based models (Agrawal et al., 2013; Prabhu and Varma, 2014; Jain et al., 2016), sparse linear models (Babbar and Schölkopf, 2017, 2019; Yen et al., 2017), and embedding-based neural networks (Guo et al., 2019), are also promising model candidates for ANN search since they are specifically tailored for multilabel classification problems with extremely large label spaces.

Our formulation enables analyzing ANN search in the statistical learning framework, thus opening multiple theoretical research questions: (1) Can we establish a sufficient condition for *strong* consistency? (2) Can we prove consistency of more adaptive partitioning classifiers, such as PCA trees or classification trees? (3) Can we establish faster than logarithmic convergence rates? The last question is especially interesting, since prediction times of trees are logarithmic: a positive answer would theoretically justify decreasing query times by increasing the training set size.

## Acknowledgements

Funding in direct support of this work: Academy of Finland grants #345635 (DAISY), #311277 (TensorML), and #313857 (WiFiUS). The authors wish to thank the Finnish Computing Competence Infrastructure (FCCI) for supporting this project with computational and data storage resources.

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
