# OpenReview forum: "A Multilabel Classification Framework for Approximate Nearest Neighbor Search"
_NeurIPS.cc/2022/Conference — NeurIPS 2022 Accept_

### Official Review · Reviewer_jQnM · 2022-06-27

**Rating:** 3
**Confidence:** 3
**Soundness:** 2 fair
**Presentation:** 2 fair
**Contribution:** 2 fair

**Summary:**

This paper focuses on the approximate nearest neighbor (ANN) search problem. The main contribution is that authors formulate candidate set selection in ANN search directly as a multilabel classification problem where the labels correspond to the nearest neighbors of the query point, and interpret the partitions as partitioning classifiers for solving this task.

**Questions:**

I have listed some problems above. It is suggested to focus on the key contributions when writing the manuscript. Moreover, the resulted problem has large label space, maybe the emerging learning task extreme multi-label learning/classification will be useful.

**Limitations:**

Authors have listed two limitations in Section 7. However, I think that the large label space is the most limitation of the proposed framework. Moreover, the pre-processing time of computing the true nearest neighbors for training examples will be very long when the size of training set is large. Unfortunately, when we need approximate nearest neighbor search, the size of training set is usually very large. Otherwise, we can obtain the accurate nearest neighbors easily.

**Strengths And Weaknesses:**

Strengths:
1. The motivation is novel.
2. A concrete implementation is provided.
3. Some experimental results are provided.

Weaknesses:
1. The motivation might be unreasonable. The size of label space equals the number of training examples which is usually very large in approximate nearest neighbor (ANN) search problem. For example, the benchmark data set GIST has one million training examples, it is very difficult to learn a multi-label classifier with so many labels.
2. This paper is not well organized. Key contributions are introduced very briefly while related works are demonstrated too detailedly. For example, the proposed method is only introduced with less than half of one page (Section 3). For another example, the experimental setting is not introduced clearly, at least the number of nearest neighbors is not given in experiments.

---

> ### Author Response · Authors · 2022-08-02
> **Author response to Reviewer jQnM**
>
> **Weakness 1.** The reviewer suggests that learning a multilabel classifier for large spaces is very difficult, and for this reason "the motivation [of our article] might be unreasonable". While it is true that the larger label spaces lead to slower model fitting times in general, there is nothing inherent in large label spaces *per se* that makes learning a multilabel classifier intractable. When tree-based (or more generally, partition-based) techniques, such as the ones we discuss in the paper, are used for multilabel classification, at each node we have to explicitly handle only the labels that appear in the training data points of that node; this is typically only a small subset of the whole label space.
>
> Indeed, in our article we demonstrate in practice that a simple classification tree can be easily adapted into this multilabel learning problem and that the training times are reasonable  (cf. Appendix D.1. for index construction times). We also emphasize that we apply the pick-all-labels (PAL) reduction to simplify the multilabel classification problem. The PAL reduction enables handling all the labels by a single multiclass classification tree. Thus, the multilabel approach does not necessarily require learning $m$ binary classifiers, where $m$ is the size of the label space, as in the traditional one-versus-all (OVA) reduction; that would indeed be very inefficient for the large label spaces.
>
> **Weakness 2.** It is true that the structure of the paper can be improved (as Reviewer wPJV also pointed out), since some of the analytical content is now misleadingly located in the related work section (Sec. 4). However, our key contributions are not restricted to Sec. 3. We would like to emphasize that the aim of this paper is not to propose a single algorithm for ANN search. Rather, we propose a widely applicable theoretical framework which immediately suggests a meta-algorithm (the natural classifier introduced in Sec. 2.3) that can be used in combination with any partition-based index structure to improve its ANN retrieval performance compared to the earlier lookup-based candidate set selection methods (as discussed in Sections 2.3 and 4). We also consider the consistency results of Sec. 5 among our key theoretical contributions.
>
> We would also like to point out that we give the number of the nearest neighbor searched for ($k=10$) on the line 286, and refer to the first two paragraphs of Sec. 6 and Appendix B for the further details of the experimental setup. In case some key details are still missing, we would be happy to make any further clarifications.
>
> **Limitations.** The reviewer questions the scalability of our approach by suggesting that the preprocessing time required to compute the true nearest neighbors of the training set points will be very long when the size of the training set is large. This is a valid concern, and we indeed explicitly mention it in the section on the limitations of our approach. However, see our comment concerning **Weakness 1**  above: the methods we consider can easily scale up to millions of labels; furthermore, one way to speed up preprocessing is to approximate nearest neighbors of the training set points and use these approximate neighbors as their labels for training. According to our initial experiments (cf. Appendix D.4.), even 50% label noise can be allowed without significantly degraded performance.

---

### Official Review · Reviewer_wPJV · 2022-07-12

**Rating:** 8
**Confidence:** 3
**Soundness:** 4 excellent
**Presentation:** 3 good
**Contribution:** 4 excellent

**Summary:**

This paper revisits the space-partitioning-based methods of approximate nearest neighbor search and proposes a new multilabel classification framework. Space-partitioning-based ANN search algorithms outputs a candidate list of $k$-nearest neighbors based on an index structure. This paper views the candidate set selection problem as a multilabel classification problem, by regarding the retrieval of the indices of the candidates as marking the query point with $k$ labels. Given a multilabel classifier, its use as an ANN algorithm is straightforward. The paper also establishes a sufficient condition for a partitioning-classifier-based ANN search algorithm, and shows that the chronological kd tree is asymptotically consistent when it is used for ANN search. Empirical evidences support that the proposed framework is indeed powerful.

**Questions:**

- While it is explained, in line 298, that "*as expected based on the theory*, the performance of partition-based index structures can indeed be improved by interpreting them as partitioning classifiers," I do not see which "theory" a priori predicts that the natural classifier would outperform the naive classifier in the proposed framework. Is it based on the observation that the Bayes classifier in the form of a plug-in classifier which is the form of the natural classifier, while the naive classifier is of a different form? I would appreciate if the authors can elaborate this.


**Limitations:**

Limitations are well discussed.

**Strengths And Weaknesses:**

**Strengths**
- The idea of viewing the candidate set selection problem as a multilabel classification problem is natural and neat. In particular, it suggests that the existing space-partitioning based methods are based on a possibly suboptimal classifier (which they call the naive classifier) and provides a simple wrapper to improve the previous unsupervised space-partitioning methods by constructing a better conditional probability estimate. Further, since this framework allows to plug-in any classifier, it opens up a question whether and when a learned classifier can outperform the unsupervised methods.
- The presentation is very clear (except one suggestion below).
- The experiments are well elucidating the ideas of the paper.

**Weaknesses**
- One minor complain is that the current flow is a bit misleading. I would expect to see Section 3 right after Sections 2.1 and 2.2, as the multilabel classification framework is independent of the form of classifiers. After Section 3, the authors can explain partitioning classifiers, the notion of natural classifiers, and the comparison with the existing lookup search based methods (essentially the content in Section 4.) The current flow does not prevent understanding the ideas by much, but I believe that the perspective of naive and natural classifiers and the discussion in Section 4 deserve better presentation, considering that it is the main empirical improvement shown in Section 6 (Figure 1).

---

> ### Author Response · Authors · 2022-08-02
> **Author response to Reviewer wPJV**
>
> The suggestion about the more logical order of the sections is very good. We will incorporate it into the manuscript.
>
> **Question 1.** "As expected based on the theory" is indeed a bit too strong wording; we will revise this. The interpretation suggested by the reviewer is basically correct: since the natural classifier thresholds the maximum likelihood estimates (under the widely used pick-all-labels and one-versus-all multilabel problem reductions) for the conditional label probabilities, and the Bayes classifier is also obtained by thresholding these conditional label probabilities, it seems plausible that this classifier is more accurate compared to the naive classifier that is of a different form (the naive classifier can actually be interpreted as a natural classifier for a different multilabel learning problem as discussed in Sec. 4).
>
> We actually briefly motivate the natural classifier from the maximum likelihood perspective in the beginning of Appendix C. We will move this discussion into the main article if the space limit allows it.

---

### Official Review · Reviewer_efJe · 2022-07-23

**Rating:** 4
**Confidence:** 5
**Soundness:** 2 fair
**Presentation:** 3 good
**Contribution:** 1 poor

**Summary:**

This paper tried to formulate the selection of candidate set of approximate nearest neighbor (ANN) search as a multilabel classification problem, and proposed a theoretical framework with experiments to show the effectiveness of "recall vs. query time" tradeoff.

**Questions:**

1) Why this paper does not consider the evaluation criteria of "precision" or "F-score" that are very common to evaluate the search quality of ANN?

2) Why this paper does not discuss the high-dimensional side-effects regarding the well-know "curse of dimensionality"?

3) Why the re-formulation of ANN search is needed to do in such a multilabel classification to find the candidate set of an ANN search?

**Limitations:**

The limitations of the proposed framework have been fairly addressed in the paper.

**Strengths And Weaknesses:**

* Strengths

1) Quality: high.
The re-formulation of ANN search as a multilabel classification is very theoretical with sufficient mathematical proofs in detail but as appendices. In terms of quality, the theoretical details are very high to show that the authors are trying to guarantee the correctness of all technical contents. However, such a theoretical way oppositely re-formulated a simple ANN search problem into another form that seems complex.

2) Clarity: high.
The whole paper is well-written and well-organized, thus it is easy to follow and understand.

* Weaknesses

Regarding the originality, significance, and completeness of the proposed method, the following concerns should be carefully considered.

1) The re-formulation of ANN search as a multilabel classification is claimed by the authors that the proposed method is an intuitive theoretical framework. However, with the multilabel information that is mapped into binary vectors, the ANN search processing actually can be easily applied to any existing lookup-based indexing methods, rather than re-formulating it into a multilabel classification problem to select candidate set of a probed ANN search. Thus, the motivation (why such as re-formulation is necessary), and the originality (how the re-formulation is originally different from the existing looku-based indexing methods if only applying the multilabel binary vectors but not think about supervised or unsupervised) are not clear nor convincing to the readers.

2) The proposed framework only targets at the measurement of "average recall vs. query time" trade-off, but neglects the very important criteria including "precision" or "F-score". As we know, an ANN search can just simple return the all data points in the dataset as search result that can guarantee the recall in 100% coverage without any computations that need costly query time. After confirming the experiments and their details in supplementary materials, any evaluations in terms of precision or F-score are not conducted. Therefore, the completeness of experimental evaluations is low and insufficient to prove the real effectiveness of the proposed framework.

3) In the experiments, we can see that the dimensionality of all data sets is extremely high ($>100$), however, the authors did not discuss the impact of such high-dimensional side-effects that should dramatically downgrade the recall and query processing efficiency. Such a side-effect is the well-known "curse of dimensionality". Any additional discussions should be considered for ANN search when using high-dimensional data sets for experiments.

4) It should be good to evaluate the pre-processing time of the proposed framework by comparing to those existing lookup-based methods because they also need  heavy computational time to build the indexes before running an ANN search.

---

> ### Author Response · Authors · 2022-08-02
> **Author response to Reviewer efJe**
>
> **Weakness 1 & Question 3.** If we understand correctly, the reviewer questions the utility of reformulating the candidate set selection for ANN search as a multilabel classification problem since the existing lookup-based methods for the candidate set selection work without this reformulation. The concise answer to this question is that the proposed reformulation enables improving performance of the partition-based index structures for ANN search since it suggests a straightforward algorithm (the *natural classifier* introduced in Sec.2.3) for candidate set selection that leads to improved performance compared to the earlier lookup-based methods. In particular, in Sec. 4 we suggest that the existing candidate set selection methods (lookup and voting) are suboptimal in view of the proposed framework, and in Sec. 6 we validate this finding empirically by showing that the performance of the natural classifier is superior compared to the lookup-based methods for all the index structures that we tested.
>
> **Weakness 2 \& Question 1.**  We believe that this critique is based on a mistaken assumption on how we measure the performance of ANN algorithms in our experiments; we try to clarify the issue in what follows. The query of the partitioning-based method consists of two steps: 1) candidate set selection; and 2) finding the approximate nearest neighbors among the points of this candidate set. While it is indeed true that many earlier articles use precision (of the candidate set) to measure the performance of ANN algorithms, they use it merely as a proxy for the time taken by the step 2 (the time complexity of the step 2 is linear w.r.t. the candidate set size). In contrast, we measure the actual time taken by the query (i.e., the time taken by both of these steps), and thus our approach is actually more–not less–indicative of the real-world performance. We would also like to point out that using the actual query time instead of the precision to measure the performance of ANN algorithms is a standard practice in the benchmark articles; see e.g., Aumüller et. al. (2020).
>
>
> To give a concrete example of how the query time–recall tradeoff is not prone to trivial solutions, the hypothetical algorithm given by the reviewer that always selects all the corpus points to the candidate set would perform very poorly in our experiments, since the query time that we measure also includes the time taken by the exact search in this very large candidate set (i.e., time taken by the step 2).
>
> **Weakness 3 & Question 2.** The concerns about high-dimensional data and the "curse of dimensionality" are valid. However, in most modern applications of ANN algorithms, the data sets indeed are high-dimensional (>100 dimensions). Our choice of benchmark data sets reflects this reality and the experiments show that ANN algorithms are still extremely useful even in this setting.
>
> **Weakness 4.** Concerning preprocessing times, see Appendix D (Sections D.1 & D.4 and Tables 3 & 5). Note also that the "lookup-based methods" refer to the candidate set selection methods (lookup search and voting) and not the index structures themselves, since the same index structures (as we demonstrate using RP, PCA, and $k$-d trees) can be used in combination with different candidate set selection methods (with lookup-based methods and with the natural classifier as suggested in this article).
>
> Aumüller, M., Bernhardsson, E., & Faithfull, A. (2020). ANN-Benchmarks: A benchmarking tool for approximate nearest neighbor algorithms. Information Systems, 87, 101374.

---

### Author Response · Authors · 2022-08-02
**General author response**

We thank all the reviewers for their feedback. There were good suggestions, especially about the overall structure, that we will try to incorporate into the manuscript. It also seems that some critical remarks were based on the misunderstandings of our methodology and contributions, and we will do our best to address them separately as comments to each reviewer.

---

### Author Response · Authors · 2022-08-05
**Rebuttal revision**

We uploaded a revised version of the article where we addressed the weaknesses pointed out by the reviewers.

The most important revisions are as follows:
- We addressed Weakness 3 and Question 2 by Reviewer efJe by discussing the effect of the "curse of dimensionality" for the nearest neighbour search on the high-dimensional data to motivate why we study approximate nearest neighbor problem (Sec. 2.1 of the revised article).
- We updated the structure of the article by moving the section on the natural classifiers (Sec. 2.3 in the original submission) after the section introducing our multilabel formulation (Sec. 3 in both of the versions) and motivated the natural classifier in a more detail as suggested by Reviewer wPJV.
- We updated the structure of the article by moving the part where we showed how the natural classifier proposed in the article differs from the earlier candidate set selection methods (naive classifier) when they are interpreted in our multilabel classification framework out of the related work section (Sec. 4 in the original article) into its own section (Sec. 6 in the revised article), and added a paragraph clarifying the difference as suggested by Reviewers jQnM (Weakness 2) & wPJV.
- We updated the wording of the description of the experimental results and clarified in the beginning Sec. 6 (of the revised article) why the natural classifier seems apriori more suitable for candidate set selection for ANN search than the naive classifier as suggested by Reviewer wPJV.

---

### Meta-Review · Area_Chair_S68r · 2022-08-26

**Recommendation:** Accept
**Confidence:** Less certain

**Metareview:**

In this paper, the authors propose a novel multi-label classification framework in the spatial partitioning-based method of approximate nearest neighbor search. The space partitioning based ANN search algorithm outputs the nearest neighbor candidate list based on the index structure. In this paper,  the problem of candidate set selection to be a multi-label classification problem is considered, and searching an index of candidates as labeling query points. The paper is theoretically solid and gives accurate answers to the questions of the reviewers during the discussion period. In particular, the main point of this paper is that existing search-based algorithms can be viewed as specific sub-optimal methods in multi-label classification frameworks. The idea is original and the evaluation is appropriate.

**Award:**

No

---

### Decision · Program_Chairs · 2022-09-14

Accept